# A clean and membrane-free chlor-alkali process with decoupled $Cl_2$ and $H_2$/NaOH production

Mengyan Hou[1], Long Chen[1], Zhaowei Guo[1], Xiaoli Dong[1], Yonggang Wang [1] & Yongyao Xia[1]

Existing chlor-alkali processes generally use asbestos, mercury or fluorine-containing ion-exchange membranes to separate the simultaneous chlorine production on the anode and hydrogen production on the cathode, and form sodium hydroxide in the electrolyte. Here, using the $Na^+$ de-intercalation/intercalation of a $Na_{0.44}MnO_2$ electrode as a redox mediator, we decouple the chlor-alkali process into two independent steps: a $H_2$ production step with the NaOH formation in the electrolyte and a $Cl_2$ production step. The first step involves a cathodic $H_2$ evolution reaction ($H_2O \rightarrow H_2$) and an anodic $Na^+$ de-intercalation reaction ($Na_{0.44}MnO_2 \rightarrow Na_{0.44-x}MnO_2$), during which NaOH is produced in the electrolyte solution. The second step depends on a cathodic $Na^+$ intercalation reaction ($Na_{0.44-x}MnO_2 \rightarrow Na_{0.44}MnO_2$) and an anodic $Cl_2$ production ($Cl \rightarrow Cl_2$). The cycle of the two steps provides a membrane-free process, which is potentially a promising direction for developing clean chlor-alkali technology.

[1] Department of Chemistry and Shanghai Key Laboratory of Molecular Catalysis and Innovative Materials, Institute of New Energy, iChEM (Collaborative Innovation Center of Chemistry for Energy Materials), Fudan University, 200433 Shanghai, China. Correspondence and requests for materials should be addressed to Y.W. (email: ygwang@fudan.edu.cn) or to Y.X. (email: yyxia@fudan.edu.cn)

The chlor-alkali process plays a predominant and irreplaceable role in the chemical industry because its products are used in over 50% of all industrial chemical processes[1–11]. However, the chlor-alkali industry is among the highest energy-consuming processes with pollutant emissions that have a serious effect on the environment and human life[1–4, 11–16]. Accordingly, it is desirable to develop a more efficient and cleaner chlor-alkali process.

Typical chlor-alkali electrolysis (i.e. brine electrolysis) consists of two half reactions: the hydrogen evolution reaction (HER) on the cathode and the chlorine evolution reaction (CER) on the anode, which is accompanied by sodium hydroxide (NaOH) formation in the electrolyte. From 1888 till now, three processes emerged in the chlor-alkali industry: the diaphragm cell, mercury cell and present membrane cell[1–4, 17, 18]. In a diaphragm cell, a porous asbestos mat is used to separate the anodic CER and cathodic HER, and NaOH is simultaneously formed in the cathodic compartment (Supplementary Fig. 1)[1–4]. In the mercury cell, the CER on the anode is coupled with the formation reaction of sodium amalgam ($2Hg + 2Na^+ + 2e^- \rightarrow 2NaHg$) on the liquid mercury cathode (Supplementary Fig. 2)[1–4]. Then, the sodium amalgam is separated and used in the second cell to produce NaOH via the anodic reaction of Na-ion release ($NaHg \rightarrow Na^+ + Hg + e^-$) and the cathodic HER (Supplementary Fig. 2)[1–4]. The operation of the present membrane cell is very similar to the diaphragm cell, but the difference is to use a polymer ion-exchange membrane instead of the porous asbestos mat (Supplementary Fig. 3)[1–4]. Because of the separated formation of $Cl_2$ and NaOH, the mercury cell generally produces a purer product than do the diaphragm and membrane cells but with higher energy consumption because of the increased cell voltage for electrolysis (Supplementary Fig. 2)[1–4].

Over the past decades, many studies have been conducted to reduce the pollution related to chlor-alkali processes. The environmental measures implemented over the past several years have successfully alleviated the issue of huge mercury emissions. In the diaphragm and membrane process, the use of asbestos or fluorine-containing materials only causes indirect emissions. Because of the limited lifetime (approximately several years) of these materials, the diaphragm and membrane process only result in moderate environmental risks. Furthermore, the membrane cell has dominated the current chlor-alkali industry. The application of oxygen-depolarised cathodes in the membrane cell also reduces the electric energy consumption in the chlor-alkali process[5, 19]. However, the general applications of this membrane-based chlor-alkali process remain challenging. Although this method can facilitate the separation of the products because of its chemical resistance, the expensive ion exchange membrane generally exhibits limited useful life[20–27]. In particular, this membrane is susceptible to contaminant ions such as $Mg^{2+}$ and $Ca^{2+}$ in the brine, which shorten the useful lifetime of the membranes[20–23]. The precipitated calcium and magnesium will decrease the efficiency and increase the power consumption. In addition, the high-pressure gases in the electrolytic cell aggravate the membrane degradation[28]. Accordingly, unstable sustainable energy sources such as wind and solar energy are difficult to use to directly power the membrane cell, because the gas pressures in the anode and cathode compartments must remain in balance with a stable power input. Therefore, it is highly desired to develop an environmentally friendly, high-efficiency, membrane-free chlor-alkali process. In fact, the old mercury cell can be considered as a typical membrane-free chlor-alkali technology, where the redox mediator of amalgam/sodium amalgam (Hg/NaHg) decouples the $H_2$ (and NaOH) production and $Cl_2$ production. Unfortunately, because of the high toxicity, the mercury-cell-based chlor-alkali technology must be stopped step by step.

However, the old mercury cell gives us the inspiration that the reversible $Na^+$-storage reaction can decouple the chlor-alkali technology. In recent years, sodium-ion batteries (SIBs) are attracting extensive attention as a promising alternative candidate to lithium-ion batteries (LIBs) because of the abundant natural reserve and low cost of sodium[29–35]. The reversible $Na^+$ intercalation/de-intercalation of the electrode is also expected to be used as a redox mediator to decouple the chlor-alkali technology.

Here we report a membrane-free chlor-alkali electrolysis process, where the $Cl_2$ evolution and $H_2$/NaOH production are decoupled by the reversible Na-ion intercalation/de-intercalation reaction of the $Na_{0.44}MnO_2$ electrode. This decoupled strategy shows promise in developing a clean chlor-alkali technology.

## Results

**Mechanism of the two-step chlor-alkali electrolysis.** As shown in Fig. 1a, the electrolysis process includes a $H_2$ (+NaOH) production step (Step 1) in a NaOH solution and a $Cl_2$ evolution step (Step 2) in a saturated NaCl solution. Step 1 involves the anodic $Na^+$ de-intercalation from the $Na_{0.44}MnO_2$ electrode (Eq. 1) and the cathodic reduction of $H_2O$ on the HER electrode to produce $H_2$ and $OH^-$ (Eq. 2). After being washed with brine, the $Na_{0.44-x}MnO_2$ (i.e. desodiated $Na_{0.44}MnO_2$, which is formed in Step 1) is used as the cathode in Step 2 for the $Cl_2$ evolution, where the anodic $Cl_2$ evolution reaction on the CER electrode (Eq. 3) is coupled to the cathodic $Na^+$ intercalation in the $Na_{0.44-x}MnO_2$ electrode (Eq. 4). Steps 1 and 2 can be cycled by moving the $Na_{0.44}MnO_2/Na_{0.44-x}MnO_2$ electrode between cell 1 for Step 1 and cell 2 for Step 2 (Fig. 1a).

Step : 1

$$\text{Anode} : Na_{0.44}MnO_2 \rightarrow Na_{0.44-x}MnO_2 + xNa^+ + xe^- \quad (1)$$

$$\text{Cathode} : 2H_2O + 2e^- \rightarrow H_2 + 2OH^- \quad (2)$$

Step : 2

$$\text{Anode} : 2Cl^- \rightarrow Cl_2 + 2e^- \quad (3)$$

$$\text{Cathode} : Na_{0.44-x}MnO_2 + xNa^+ + xe^- \rightarrow Na_{0.44}MnO_2 \quad (4)$$

This approach creates a chlor-alkali electrolysis process with several important advantages. First, because of the environmentally friendly nature of $Na_{0.44}MnO_2$, this technology is cleaner than diaphragm cells, mercury cells and even the current membrane cells. Second, this architecture can separately produce $H_2$ (+NaOH) and $Cl_2$ without using a membrane. Finally, the separate generation of $Cl_2$ and $H_2$ prevents the mixing of product gases over a range of current densities and also simplifies the gas handling, which significantly increases the operational flexibility of chlor-alkali electrolysis cells and potentially make them suitable to be driven by sustainable energy sources (such as solar or wind energy).

$Na_{0.44}MnO_2$ with a three-dimensional (3D) S-shaped tunnel structure was prepared via a simple solid reaction (see the experimental section) according to previous reports[36–40]. The X-ray diffraction (XRD) pattern, scanning electron microscopy (SEM) image and transmission electron microscopy (TEM) image of the prepared sample are shown in Supplementary Figs. 4 and 5. Prior to the fabrication of this chlor-alkali electrolytic cell, the intercalation/de-intercalation behaviours of the $Na_{0.44}MnO_2$ electrode in an alkaline electrolyte (1 M NaOH) and a brine

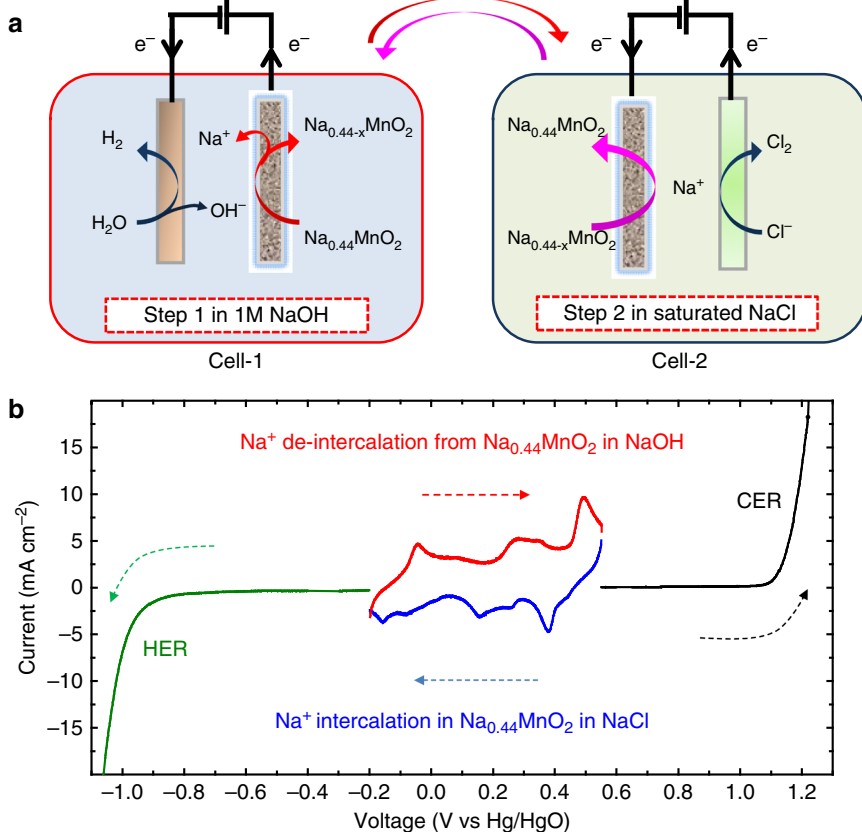

**Fig. 1** Mechanism of two-step chlor-alkali electrolysis. **a** Schematic illustration of the operation mechanism of the electrolysis cell. Step 1 involves the cathodic reduction of $H_2O$ on the HER electrode and anodic $Na^+$ de-intercalation from the $Na_{0.44}MnO_2$ electrode in an NaOH solution (Cell 1); Step 2 depends on the cathodic $Na^+$ intercalation on the $Na_{0.44-x}MnO_2$ electrode and anodic $Cl_2$ evolution reaction on the CER electrode in a saturated NaCl solution (Cell 2). **b** Linear sweep voltammetric (LSV) data at a sweep rate of $2\,mV\,s^{-1}$ of $Na^+$ de-intercalation from $Na_{0.44}MnO_2$ in 1 M NaOH (red line) and $Na^+$ intercalation in $Na_{0.44-x}MnO_2$ in a saturated NaCl solution (blue line); LSV data of the commercial Pt-coated Ti-mesh electrode for HER at a sweep rate of $2\,mV\,s^{-1}$ in 1 M NaOH solution (green line); LSV data of commercial $RuO_2/IrO_2$-coated Ti-mesh electrode for CER in a saturated NaCl solution (black line) at a sweep rate of $2\,mV\,s^{-1}$

electrolyte (saturated NaCl) were investigated using cyclic voltammetry (CV) measurements with a typical three-electrode system (Supplementary Figs. 6, 7). As shown in Supplementary Figs. 6 and 7, the prepared $Na_{0.44}MnO_2$ exhibits identical electrochemical profiles in both electrolytes, which indicates a possibility of the reversible cycle between the desodiation (i.e. $Na^+$ de-intercalation) in the alkaline electrolyte and the sodiation (i.e. $Na^+$ intercalation) in the brine electrolyte of one $Na_{0.44}MnO_2$ electrode. To further clarify this point, the desodiation of a $Na_{0.44}MnO_2$ electrode in an alkaline electrolyte and consequent sodiation of the resulting electrode in a brine electrolyte were investigated using linear sweep voltammetry (LSV) measurements with a three-electrode system at a sweep rate of $2\,mV\,s^{-1}$ (Fig. 1b). The HER on a Pt-coated titanium mesh electrode in an NaOH solution and CER on a $RuO_2/IrO_2$-coated titanium mesh electrode in a brine electrolyte were also investigated by LSV measurements for comparison (Fig. 1b). As shown in Fig. 1b (the red line), five desodiation peaks at −0.04, 0.08, 0.28, 0.34 and 0.49 V (vs. Hg/HgO) appear in the positive sweep process (from −0.2 to 0.55 V vs. Hg/HgO) in an alkaline electrolyte, and they indicate a desodiation reaction of $Na_{0.44}MnO_2 \rightarrow Na_{0.44-x}MnO_2 + xNa^+ + xe^-$. Then, the resulting $Na_{0.44-x}MnO_2$ electrode was put in the brine electrolyte for the LSV measurement with a negative sweep from 0.55 to −0.2 V vs. Hg/HgO. As shown in Fig. 1b (the blue line), the sodiation peaks in the brine electrolyte appear at 0.38, 0.25, 0.16, −0.08 and −0.16 V vs. Hg/HgO in the negative sweep process. The potential gap between the main peaks of

desodiation in the alkaline electrolyte and those of sodiation in the brine electrolyte is ~0.11 V, which indicates a good reversibility. Furthermore, the special potential window for the $Na_{0.44}MnO_2/Na_{0.44-x}MnO_2$ redox couple is between the onset potentials for the HER and CER (the green and black lines in Fig. 1b, respectively). The result indicates that $Na_{0.44}MnO_2$ can be used as a redox mediator to decouple the conventional chlor-alkali process into two steps, according to Fig. 1a. The galvanostatic charge–discharge curve of the $Na_{0.44}MnO_2$ electrode at a current density of $0.1\,A\,g^{-1}$ is shown in Supplementary Fig. 8 to clarify the specific capacity of $Na_{0.44}MnO_2$.

**Performance of the two-step chlor-alkali electrolysis.** The chlor-alkali electrolytic process in Fig. 1a was constructed with a commercial Pt-coated Ti-mesh electrode for the HER ($2.5 \times 4$ $cm^2$, Supplementary Fig. 9a), a commercial $RuO_2/IrO_2$-coated Ti-mesh electrode for the CER ($2.5 \times 4$ $cm^2$, Supplementary Fig. 9b) and an $Na_{0.44}MnO_2$ electrode ($2.5 \times 4$ $cm^2$, Supplementary Fig. 9c). According to Fig. 1a, Step 1 (NaOH/$H_2$ production) and Step 2 ($Cl_2$ production) were conducted in two separated cells with the alkaline electrolyte and brine electrolyte, respectively. The electrolysis of the cells was investigated by chronopotentiometry with an applied current of 100 mA and a step time of 600 s, and the corresponding chronopotentiometry curves (cell voltage vs. time) are shown in Fig. 2a. The chronopotentiometry data of the anode (anodic potential vs. time) and cathode (cathodic potential vs. time) were also investigated

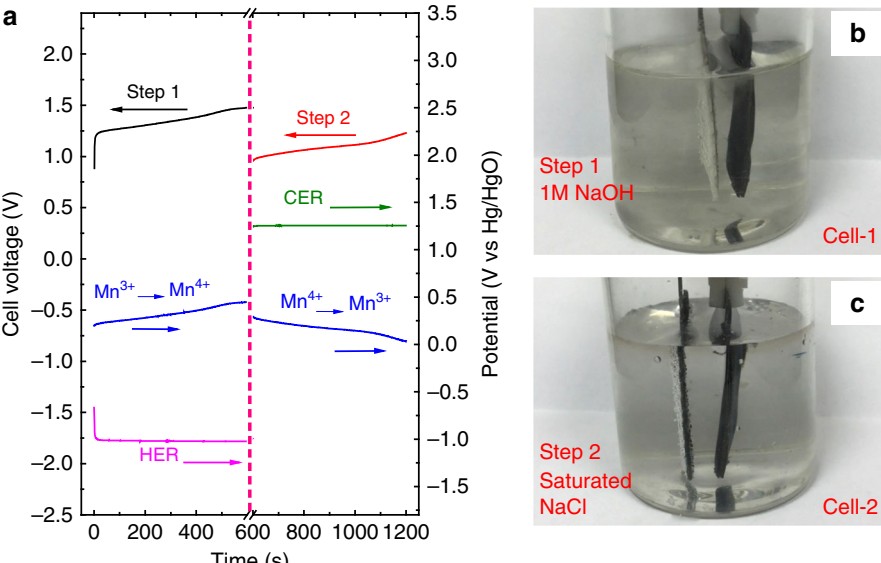

**Fig. 2** Electrochemical profile of two-step chlor-alkali electrolysis. **a** Chronopotentiometry curves (cell voltage vs. time) of Step 1 (NaOH + $H_2$ production in Cell 1) and Step 2 ($Cl_2$ production in Cell 2) at a constant applied current of 100 mA. Chronopotentiometry data (potential vs. time) of the HER electrode (pink line), CER electrode (green line) and $Na_{0.44}MnO_2/Na_{0.44-x}MnO_2$ electrode (blue line) are shown in Fig. 2a. [(Voltage of Step 1) = (Potential of $Na^+$ de-intercalation)−(Potential of HER); (Voltage of Step 2) = (Potential of CER)−(Potential of $Na^+$ intercalation)]. **b**, **c** Photo profiles of the $H_2/Cl_2$ generation in Steps 1 and 2, where $H_2$ and $Cl_2$ are produced on the HER (**b**) and CER electrodes (**c**), respectively (Supplementary Movies 1 and 2 further confirm this point)

during the electrolysis process and are shown in Fig. 2a. In Fig. 2a, Step 1 (i.e. the NaOH/$H_2$ production process) exhibits an average cell voltage of ~1.3 V, which arises from the difference between the average anodic potential of 0.3 V (vs. Hg/HgO) of the desodiation ($Na_{0.44}MnO_2 \rightarrow Na_{0.44-x}MnO_2 + xNa^+ + xe^-$) and the cathodic potential of approximately −1.0 V (vs. Hg/HgO) of the $H_2O$ reduction ($2H_2O \rightarrow H_2 + 2OH^-$). Then, after the washing with NaCl solution, the resulting $Na_{0.44-x}MnO_2$ formed in Step 1 was moved to another cell for Step 2 (the $Cl_2$ production process). The average cell voltage in Step 2 was ~1.07 V, which is equal to the average potential difference between the anodic CER ($2Cl^- + 2e^- \rightarrow Cl_2$, ~1.24 V vs. Hg/HgO) and cathodic sodiation ($Na_{0.44-x}MnO_2 + xNa^+ + xe^- \rightarrow Na_{0.44}MnO_2$, ~0.17 V vs. Hg/HgO). The photo profiles of the $H_2$ generation in Step 1 and $Cl_2$ generation in Step 2 are shown in Fig. 2b and c, respectively, to characterise the decoupled steps. Furthermore, video evidence is shown to further clarify the separated $H_2$ generation in Step 1 and $Cl_2$ generation in Step 2 (see Supplementary materials, Supplementary Movies 1 and 2). The electrolysis of the cells with a longer step time of 1 h (3600 s) was investigated by chronopotentiometry at an applied current of 100 mA (Supplementary Fig. 10). The chlor-alkali electrolytic process with the decoupled $H_2$ (+NaOH) production and $Cl_2$ production was further characterised by the chronopotentiometry measurements at the higher applied currents of 200, 500 and 1000 mA, respectively (Supplementary Fig. 11). The corresponding video proofs for the electrolysis at 1000 mA are shown in Supplementary Movies 3 and 4. When tested at a high current of 1000 mA (= current density of 100 mA $cm^{-2}$), the overall cell voltage (average voltages on Step 1 + Step 2) is ~3.65 V. Obviously, the applied current density (100 mA $cm^{-2}$) is still much lower than the typical current densities in modern membrane cells (150–700 mA $cm^{-2}$)[8]. For this case, the electrolysis rate in the cell is limited by the $Na^+$ intercalation/ de-intercalation in the crystalline framework of solid electrode material ($Na_{0.44}MnO_2$). An efficient solution for this issue is to reduce the current density (mA $cm^{-2}$) applied on the $Na_{0.44}MnO_2$ electrode, which can be achieved by increasing the area ($cm^2$)

ratio between the $Na_{0.44}MnO_2$ electrode and the CER (or HER) electrode. For example, a higher applied current density of 500 mA $cm^{-2}$ on the CER or HER electrode has been successfully achieved using this method (Supplementary Fig. 12). The kinetics of the $Na^+$ or $Li^+$ intercalation/de-intercalation also controls the power performance of current rechargeable Na-ion or Li-ion batteries. Various approaches such as nanosizing and nanostructuring electrode materials have been developed to improve the power of these rechargeable batteries[41], which can also be used to improve the electrolysis rate of the new chlor-alkali process. According to the design in Fig. 1a, the $H_2$/NaOH generation (Step 1) and $Cl_2$ generation (Step 2) are performed in two separated cells: Cell 1 and Cell 2. In theory, both steps can be conducted in a single cell with the brine electrolyte (Supplementary Fig. 13a). However, the generated NaOH is dissolved in the brine electrolyte solution, which indicates that no pure NaOH stream is produced. Furthermore, with the increase of generated NaOH in the brine solution, the oxygen evolution reaction (OER) occurs on the CER electrode (Supplementary Fig. 13b). The separated cells can efficiently prevent the undesired issues, which is one of the purposes of the design in Fig. 1a. A conceptual design of the continuous production of the decoupled process is shown in Supplementary Fig. 14, and the technology challenges are briefly analysed in the corresponding discussion. According to recent reports about decoupled water electrolysis[42–45], we can assume that the decoupled chlor-alkali process can be driven by an unstable renewable energy, which may improve the cost and efficiency of electrolysers.

As a typical electrode material for sodium-ion batteries, $Na_{0.44}MnO_2$ exhibits the high efficiency and long cycle capability, which were demonstrated in a previous report on a $Na_{0.44}MnO_2$-based aqueous battery[39]. To further clarify this point, the cycle performance of $Na_{0.44}MnO_2$ was investigated using a galvanostatic charge (in the 1 M NaOH solution)/discharge (in the saturated NaCl solution) test (Supplementary Fig. 15). Obviously, the highly reversible $Na^+$ intercalation/de-intercalation behaviour of $Na_{0.44}MnO_2$ facilitates the cycle of NaOH/$H_2$ production (Step

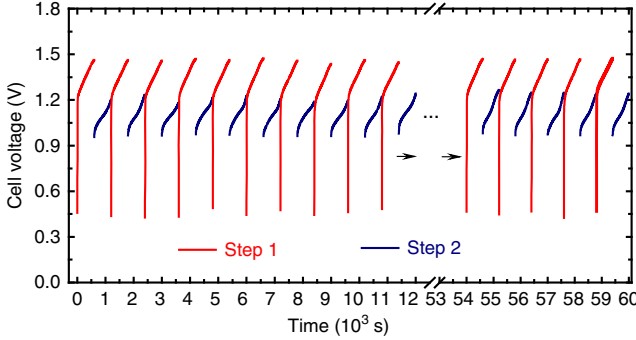

**Fig. 3** Cycle profile of two-step chlor-alkali electrolysis. Chronopotentiometry curve (cell voltage vs. time) of the cycle between Steps 1 and 2 with an applied current of 100 mA and a step time of 600 s, where the chronopotentiometry data of Step 1 ($H_2$ + NaOH generation) and Step 2 ($Cl_2$ generation) are labelled with the red and blue lines, respectively

1) and $Cl_2$ production (Step 2). Accordingly, the cycle profile of NaOH/$H_2$ production and $Cl_2$ production was investigated with an applied current of 100 mA. The achieved result is shown in Fig. 3, where the new chlor-alkali electrolytic process stably produced $H_2$ (+NaOH) and $Cl_2$ in over 50 consecutive cycles. For practical application, it is also necessary to clarify the durability of the $Na_{0.44}MnO_2$ electrode and its electrochemical stability in NaOH solution with higher concentrations. Therefore, the $Na^+$ intercalation/de-intercalation behaviour of the $Na_{0.44}MnO_2$ electrode in the 10 M NaOH (30 wt %) solution was investigated by CV measurement at a scan rate of 2 mV s$^{-1}$ (Supplementary Fig. 16), where the achieved electrochemical profile is identical to that in 1 M NaOH. This result demonstrates the high stability of $Na_{0.44}MnO_2$ in high-concentration alkaline solutions. Then, the 10 M NaOH solution was used to perform Step 1 for NaOH and $H_2$ production (Supplementary Fig. 17). The electrochemical profile of Step 1 in 10 M NaOH is identical to that in 1 M NaOH.

**Gas analysis of the $H_2$ evolution and $Cl_2$ evolution**. As shown above, the new chlor-alkali electrolytic process involves the decoupled $H_2$ (and NaOH) production step and $Cl_2$ production step. To further confirm this point, in situ differential electrochemical mass spectrometry (DEMS) was used to measure the $H_2$ evolution in Step 1 and $Cl_2$ evolution in Step 2 with a constant applied current of 100 mA. In this experiment, a quadrupole mass spectrometer with a leak inlet was connected to the chlor-alkali electrolytic cell (the cell for Step 1 or Step 2) with two tubes as the purge/carrier gas inlet and outlet (see Methods and Supplementary Fig. 18 for details). A pure Ar gas stream was used as the purge gas before the electrolysis and the carrier gas during the electrolysis process. Prior to the gas analysis, the system was purged with a pure Ar stream for 1.5 h. Then, the system was purged with a pure Ar stream for an additional hour, and an online analysis record shows that $H_2$ (Fig. 4a) or $Cl_2$ (Fig. 4b) reached a stable background line. As shown in Fig. 4a, c, when the $H_2$-production step (Step 1) begins, the $H_2$ evolution is clearly observed in the online analysis record. In parallel with the $H_2$ production, the NaOH is produced in the electrolyte solution because of the OH$^-$ generation in the HER at the cathode (Eq. 2) and $Na^+$ de-intercalation from the $Na_{0.44}MnO_2$ anode (Eq. 1). After Step 1, a rest step of 130 min was performed with a pure Ar stream to eliminate the remnant $H_2$ in the system, and a hysteresis of $H_2$ was observed in the online analysis record (Fig. 4a). Then, the resulting $Na_{0.44-x}MnO_2$ electrode that was formed in Step 1 was washed with NaCl solution to remove the surface-

adsorbed NaOH. The washed $Na_{0.44-x}MnO_2$ electrode was moved to another cell for the $Cl_2$ production (Step 2) and the corresponding DEMS analysis. As shown in Fig. 4b, d, the $Cl_2$ evolution is clearly observed in the online analysis record of DEMS during Step 2. However, it should be noted that the DEMS result shown in Fig. 4b cannot demonstrate the pure $Cl_2$ production. It is almost impossible to fully avoid the parasitic OER reaction during the CER process[8]. This point will be further clarified by the yield analysis for $H_2$ and $Cl_2$. Furthermore, the $Cl_2$ gas, which is generated at the anode, experiences immediate hydrolysis as follows: $Cl_2 + H_2O \rightarrow HClO + HCl$[46]. As a result, only a part of the generated $Cl_2$ gas is recorded by DEMS, which will be later confirmed. Step 1 is performed in an alkaline electrolyte (NaOH solution). Therefore, in theory, the NaOH generated in Step 1 is not mixed with other salts. However, the $Na_{0.44}MnO_2$ electrode is moved between the NaOH electrolyte in Step 1 and the brine electrolyte in Step 2, which can result in the presence of OH$^-$ (or Cl$^-$) in the brine electrolyte (or NaOH electrolyte). Therefore, additional washing is necessary to remove the surface-adsorbed OH$^-$ (when moving from Step 1 to Step 2) or Cl$^-$ (when moving from Step 2 to Step 1), which is one of the disadvantages of the new chlor-alkali electrolytic process. This disadvantage is further clarified by the continuous production design shown in the discussion about Supplementary Fig. 14.

A typical drainage method (Supplementary Fig. 19a) was used to quantify the $H_2$ or $Cl_2$ generation over a specific time length. In this experiment, the produced $H_2$ gas volume (mL) was measured with an applied current of 200 mA for 500 s (Supplementary Fig. 19b). The Faradaic efficiency was calculated from the ratio between the measured and theoretical $H_2$ gas volumes. Supplementary Table 1 summarises the $H_2$ production data obtained from five repeated experiments, where the Faradaic efficiency of the $H_2$ production is ~100%. With the identical measurement condition, the obtained $Cl_2$ gas volume is lower than that of $H_2$ gas because some $Cl_2$ that was generated at the anode undergoes hydrolysis ($Cl_2 + H_2O \rightarrow HClO + HCl$). The HClO and $Cl_2$ in the electrolyte were determined using the iodometric titration method (see Methods)[47]. The total chlorine that evolved because of the electrolysis of NaCl solution was determined by adding the chlorine in gas phase to the 'available chlorine' in liquid phase. The results of five repeated experiments, which are shown in Supplementary Table 2, indicate the average Faradaic efficiency of 90.2% for the $Cl_2$ production, which is lower than that for $H_2$ production. Furthermore, an ion-exchange membrane was used to evaluate the $H_2$ production and $Cl_2$ production with repeated experiments. In these experiments, the $Na_{0.44}MnO_2$ electrode and HER or CER electrode were separated by the membrane (Nafion film). The corresponding data from five repeated experiments are provided in Supplementary Table 3. The Faradaic efficiency of $H_2$ production in the membrane cell is ~100%, which is as same as that achieved without membrane. It can be detected that the calculated Faradaic efficiency (97.4%) of $Cl_2$ production with the use of membrane is still lower than 100%, indicating the presence of the parasitic OER reaction.

**Discussion**
In summary, the $Na^+$ intercalation/de-intercalation reaction has been successfully used to decouple the $H_2$ (+NaOH) and $Cl_2$ production in the conventional chlor-alkali electrolysis technology. The separated $H_2$ (+NaOH) and $Cl_2$ production in different compartments facilitates the membrane-free chlor-alkali process. This technology should be cleaner than the previous chlor-alkali electrolytic process because $Na_{0.44}MnO_2$ is environmentally friendly. Finally, as a typical Na-storage electrode material with low cost, $Na_{0.44}MnO_2$ can be easily produced on a large scale

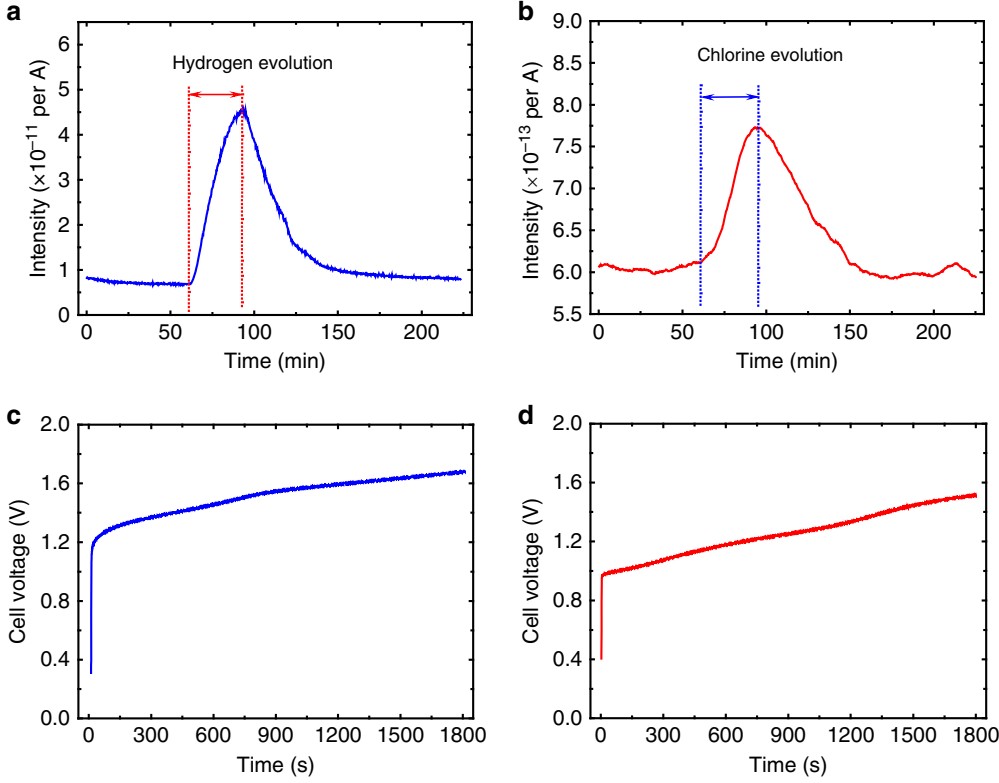

**Fig. 4** In situ differential electrochemical mass spectrometry of $H_2$ and $Cl_2$ production. **a** DEMS curve of the $H_2$ evolution and the **c** corresponding chronopotentiometry curve (cell voltage vs. time) at an applied current of 100 mA. **b** DEMS curve of the $Cl_2$ evolution and the **d** corresponding chronopotentiometry curve (cell voltage vs. time) at an applied current of 100 mA. A pure Ar gas stream was used as the purge gas before the electrolysis and the carrier gas in the total electrolysis process

when using the conventional solid-state method. This work may open the door to build a cleaner chlor-alkali electrolysis technology, where the reversible $Na^+$ storage in the electrode plays the role of a redox mediator to decouple the $H_2$ (+NaOH) and $Cl_2$ production. For example, the reversible $Na^+$ adsorption/desorption in the porous carbon electrode of electrochemical double-layered capacitors can be used to decouple the conventional chlor-alkali electrolysis (Supplementary Figs. 20, 21 and Supplementary Movies 5, 6). Furthermore, the recent chlorine technology with oxygen-depolarised cathodes[5,19] can be decoupled by the reversible $Na^+$ storage in an $Na_{0.44}MnO_4$ electrode (Supplementary Fig. 22). However, in the current state, the decoupled chlor-alkali process is still some way away from practical application. The movement of the battery electrode between Steps 1 and 2 and the corresponding washing processes requires a complex automation operation and advanced industrial design. The kinetics of the battery electrode reaction is still lower than that of the HER and the CER, so the advanced electrode is required to reduce the difference in kinetics. Other issues, such as the reachable concentrations, purity of products, required amount of $Na_{0.44}MnO_2$, parasitic OER reaction etc, also need to be solved in future research endeavours.

## Methods

**Synthesis and characterisation of $Na_{0.44}MnO_2$**. $Na_{0.44}MnO_2$ was prepared using a solid-state reaction method with $Na_2CO_3$ (99%) and $Mn_3O_4$ (99.95%) as the precursor mixture. In the typical synthesis, stoichiometric amounts of $Na_2CO_3$ and $Mn_3O_4$ were ground by ball milling (Fritsch, Pulverisette 5) at 300 RPM for 5 h. The final products were obtained by sintering the resulted mixture in a muffle furnace at 775 °C for 10 h. Powder X-ray diffraction (XRD) patterns were collected using an X-ray diffractometer (D8 Advance, Bruker), which was equipped with Cu-Kα in the 2θ range of 10–60°. The size and morphology of the powdered samples

were observed using FE-SEM (JSM-6390, JEOL) and field emission HRTEM (JEM-2010, JEOL).

**Electrode fabrication and electrochemical test**. The $Na_{0.44}MnO_2$ electrode was prepared according to the following steps. The active materials (70 wt %) were mixed well with acetylene black (8 wt %), multi-walled carbon nanotubes (8 wt %) and polytetrafluoroethylene binder (14 wt %) in isopropanol to form a homogeneous slurry. The slurry mixtures were treated with a roll press machine to form films. The films were dried in a vacuum oven at 80 °C for 12 h to remove the remaining solvent before pressing onto a titanium grid, which served as a current collector. The electrochemical profiles of the prepared $Na_{0.44}MnO_2$ electrode in an alkaline electrolyte (1 M NaOH) and a brine electrolyte (saturated NaCl) were investigated using cyclic voltammogram (CV) and galvanostatic charge–discharge measurements, respectively. The desodation (i.e. $Na^+$ de-intercalation) of the $Na_{0.44}MnO_2$ electrode in an alkaline electrolyte and consequent sodiation (i.e. $Na^+$ intercalation) of the resulting electrode in the brine electrolyte were also investigated using LSV at a sweep rate of 2 mV s$^{-1}$. The onset potential of the HER on a commercial Pt-coated Ti-mesh electrode in an alkaline electrolyte (1 M NaOH) and the onset potential of the CER on a commercial $RuO_2/IrO_2$-coated Ti-mesh electrode in a brine electrolyte (saturated NaCl) were investigated using LSVs. The aforementioned experiments were performed with a typical three-electrode method, where a Pt plate and Hg/HgO (0.098 V vs. NHE) were used as the counter and reference electrodes, respectively. All electrochemical measurements were performed on a PARSTAT MC multi-channel workstation (Princeton). In these experiments, the mass loading of $Na_{0.44}MnO_2$ in electrode was ~7 mg cm$^{-2}$.

**Fabrication of the electrolytic cell**. The cell was constructed with a commercial Pt-coated Ti-mesh electrode (Supplementary Fig. 9a) for the HER, a commercial $RuO_2/IrO_2$-coated Ti-mesh electrode for the CER (Supplementary Fig. 9b) and a $Na_{0.44}MnO_2$ electrode (Supplementary Fig. 9c; the electrode preparation is provided in the discussion of Supplementary Fig. 9c). The HER (+NaOH formation) and CER were conducted in two separated cells, which were filled with 1 M NaOH and saturated NaCl, respectively. First, the $Na_{0.44}MnO_2$ electrode (2.5 × 4 cm$^2$) was fixed by a platinum electrode cramp for the electrolysis (i.e. Step 1) in the NaOH solution. After the HER, the $Na_{0.44}MnO_2$ electrode was washed and transferred to the NaCl solution for the electrolysis of Step 2.

**Chlor-alkali electrolysis investigation**. The chlor-alkali electrolysis with two decoupled steps (i.e. Step 1 in cell 1 and Step 2 in cell 2) was investigated using chronopotentiometry measurements with applied currents of 100 mA. Step 1 was performed in cell 1 with 1 M NaOH electrolyte, in which the HER electrode (that is, Pt coated Ti-mesh electrode) and $Na_{0.44}MnO_2$ electrode were connected to the cathode and anode, respectively, of the power source for electrolysis. The duration time of Step 1 was 600 s with an applied current of 100 mA. Then, the $Na_{0.44-x}MnO_2$ electrode that was formed in Step 1 in cell 1 was washed with an NaCl solution to remove the adsorbed NaOH on the surface. The washed $Na_{0.44-x}MnO_2$ electrode was put into cell 2 with saturated NaCl for Step 2. In Step 2, the $Na_{0.44-x}MnO_2$ electrode and CER electrode (i.e. the $RuO_2/IrO_2$-coated Ti-mesh electrode) were connected to the cathode and anode, respectively, of the power source for the electrolysis. The cell voltages (V vs. time) of Steps 1 and 2 were recorded to characterise the electrolysis profile. With an additional reference electrode (i.e. the Hg/HgO electrode), the chronopotentiometry data (potential vs. time) of a single electrode (the HER electrode, $Na_{0.44}MnO_2/Na_{0.44-x}MnO_2$ electrode or CER electrode) were recorded in Steps 1 and 2. A PARSTAT MC multi-channel workstation was used to perform the chlor-alkali electrolysis investigation. The information about gas analysis is shown in Supplementary Fig. 18 and Supplementary Table 1–3.

**Data availability**. Data supporting the findings of this study are available within the Article and its Supplementary Information file, and from the corresponding author upon request.

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

## Acknowledgements

We acknowledge funding support from the Natural Science Foundation of China (21622303, 21333002), National Key Research and Development Plan (2016YFB0901500) and Shanghai Science & Technology Committee (08DZ2270500).

## Author contributions

Y.W. conceived this idea and designed the experiments. Y.W. and Y.X. directed the project. M.H. carried out the experiments. L.C., X.D. and Z.G. assisted M.H. in making some material characterisations. M.H. and Y.W. co-wrote the paper. All authors discussed the results and commented on the manuscript.

## Additional information

**Competing interests:** The authors declare no competing financial interests.

