## [Peer Review File · Nature Communications]

Reviewers' comments:

Reviewer #1 (Remarks to the Author):

Dear authors, I have now gone through your answers to my previous comments and I find the manuscript considerably improved, in particular regarding the language and the current efficiency results. My recommendation now is to publish the paper. Good luck with further studies of this interesting process.

Reviewer #5 (Remarks to the Author):

The paper deals with interesting topic new route of chlor-alkali production. The proposed electrochemical process combines MnO₂ based electrodes from sodium batteries and mercury chlor-alkali process.

Despite I found this idea interesting the self-approration of authors without pointing out several drawbacks of their process makes the paper more like advertisement than serious scientific paper. Significantly low intensity of process in contrast to the standard technologies is only mentioned. Reachable concentrations, purity of products, required amount of Na_{0.44}MnO₂ for some relevant production etc. should be mentioned as problematical point necessary to be solved prior conclusion about new technology.

Beside this general remark I have some specific comments:

In Supplementary Figure 2 the decomposer reactor is shown with power source in the circuit. The decomposer is short-circuited galvanic cell, therefore it doesn't need any external source of energy.

Declared Cl₂ production issues is hardly believable. In membrane and mercury process the brine is acidified by HCl addition. Main reason for it is suppression of parasitic oxygen production. Due to the thermodynamic preference it is impossible to fully omit oxygen evolution reaction (OER) (see reference 8). Therefore declared analysis about only pure Cl₂ formation in saturated NaCl will be real break through in chlor-alkali technology (Page 9 Figs. 4.).

To call Cl₂ production with Na intercalation as efficient approach for sea water desalination is pure speculation without any relevance (Page 9 line 235)

The discussion about current yields for Cl₂ productions is incorrect (page 10 line 257-272). If the reason for lower Cl₂ yield is back reduction on cathode how it is possible to reach 100% yield for H₂ production. Some charge must be consumed for back reduction. With Nafion membrane incorporation the penetration of oxidized chlorine compounds to the cathode is prevented. Therefore the observed yield for Cl₂ only confirms presence of OER parasitic reaction.

Reference 41 is not in right format.

Due to the mentioned reasons I recommend this paper for publication after major revision.

Response to Reviewer #5

Overall Comment: The paper deals with interesting topic new route of chlor-alkali production. The proposed electrochemical process combines MnO_2 based electrodes from sodium batteries and mercury chlor-alkali process. Despite I found this idea interesting the self-approration of authors without pointing out several drawbacks of their process makes the paper more like advertisement than serious scientific paper. Significantly low intensity of process in contrast to the standard technologies is only mentioned. Reachable concentrations, purity of products, required amount of $\text{Na}_{0.44}\text{MnO}_2$ for some relevant production etc. should be mentioned as problematical point necessary to be solved prior conclusion about new technology.

Response: Thank you very much for your kindly reviewing our manuscript (NCOMMS-17-20852A) and giving a lot of important and reasonable suggestions. These problematical points (e.g. the reachable concentrations, purity of products, required amount of $\text{Na}_{0.44}\text{MnO}_2$ for some relevant production, side reaction of OER, etc.) necessary to be solved prior the application of the new technology have been mentioned in the discussion section of the revised manuscript. **In discussion section of the revised manuscript, it is also emphasized that this idea is currently far away from the practical application, and the further research is necessary. Please see these sentences highlighted by yellow background in the discussion section (page 11 of the revised manuscript).** Furthermore, we believe your specific comments are very reasonable, which are really important to increase the quality of this work. In corresponding to each specific comment pointed out by you, we would like to answer separately and revise our manuscript according to your comments.

(1) In Supplementary Figure 2 the decomposer reactor is shown with power source in the circuit. The decomposer is short-circuited galvanic cell, therefore it doesn't need any external source of energy.

Response: As correctly pointed out by you, it doesn't need any external source of energy. Supplementary **Figure 2** has been revised carefully.

(2) Declared Cl_2 production issues is hardly believable. In membrane and mercury process the brine is acidified by HCl addition. Main reason for it is suppression of parasitic oxygen production. Due to the thermodynamic preference it is impossible to fully omit oxygen evolution reaction (OER) (see reference 8). Therefore declared analysis about only pure Cl_2

formation in saturated NaCl will be real break through in chlor-alkali technology (Page 9 Figs. 4.).

Response: Thank you very much for your kind comment. After carefully reading the nice paper (ref. 8), we have understood that it is impossible to fully omit oxygen evolution reaction (OER) during the chlorine evolution reaction (CER). The achieved data about DEMS (Figure 4b) can only be used to demonstrate the chlorine evolution, rather than the purity. We have revised the corresponding discussion carefully. **Please see the revised manuscript, Page 9, these sentences highlighted by yellow background.**

(3) To call Cl₂ production with Na intercalation as efficient approach for sea water desalination is pure speculation without any relevance (Page 9 line 235)

Response: According to your kind suggestion, the related sentences have been deleted in the revised manuscript.

(4) The discussion about current yields for Cl₂ productions is incorrect (page 10 line 257-272). If the reason for lower Cl₂ yield is back reduction on cathode how it is possible to reach 100% yield for H₂ production. Some charge must be consumed for back reduction. With Nafion membrane incorporation the penetration of oxidized chlorine compounds to the cathode is prevented. Therefore the observed yield for Cl₂ only confirms presence of OER parasitic reaction.

Response: Sincerely thank you for your kind comment. After carefully reading your comment and reference 8, we totally agree with your opinion that the observed yield for Cl₂ confirms the presence of OER parasitic reaction. Therefore, the corresponding discussion has been revised to clarify this point. **Please see the revised manuscript, Page 10, these sentences highlighted by yellow background.**

(5) Reference 41 is not in right format.

Response: Thank you very much for your comment. Reference 41, which is a website, is provided by one of the previous reviewers to clarify the typical conditions for current chlor-alkali production in membrane process. Unfortunately, the website (i.e., reference 41) currently does not work. The similar content about the typical conditions for current chlor-alkali production is also summarized in Table 1 of reference 8. Accordingly, in the revised manuscript, the corresponding citation has been replaced by reference 8. **Please see the**

revised manuscript, Page 7, the citation highlighted by yellow background.

REVIEWERS' COMMENTS:

Reviewer #5 (Remarks to the Author):

Authors reflected all my comments and recommendations. Therefore, in current version it is acceptable for publication.

Reviewer #5 (Remarks to the Author):

Reviewers' Comment: Authors reflected all my comments and recommendations. Therefore, in current version it is acceptable for publication.

Response: Thank you very much for reviewing and recommending publication of our manuscript.